# Brief Communication: Bridging the Data Gap − A Call to Enhance the Representation of Global Coastal Flood Protection

Nicole van Maanen[1]*, Joël J.-F. G. De Plaen[1]*, Timothy Tiggeloven[1], Maria Luisa Colmenares[1], Philip J. Ward[1,2], Paolo Scussolini[1] and Elco Koks[1]

[1] Institute for Environmental Studies, Vrije Universiteit Amsterdam, Amsterdam, 1081 HV, The Netherlands
[2] Deltares, Delft, 2629 HV, The Netherlands

*Correspondence to*: Nicole van Maanen (n.a.m.van.maanen@vu.nl) and Joël J.-F. G. De Plaen (joel.deplaen@vu.nl)

* Shared first-authorship

**Abstract.** Understanding coastal flood protection is crucial for assessing risks from natural hazards and climate change. However, there is a significant lack of quantitative data on coastal flood protection and their standards, posing a major barrier to risk assessment. FLOPROS, currently the only global database of flood protection standards, relies on limited coastal observations and simplified assumptions. While widely used, it cannot adequately constrain uncertainties in risk estimates that are based on it. To address this gap, we call for a global, community-driven effort to develop a more comprehensive dataset. As a first step, we present a dataset compiling COASTtal flood PROtection Standards within EUrope (COASTPROS-EU), elaborated from a survey distributed to flood practitioners from several European countries. This highlights the need for more extensive and coordinated data collection efforts, using a transdisciplinary community-based approach that ensures diverse societal representation.

## 1. Coastal Flood Protection

In Europe alone, damage from coastal flooding currently amounts to €1.4 billion annually, with around 100,000 people exposed (Feyen et al., 2020). The rise in global temperatures caused by anthropogenic greenhouse gas emissions means that the frequency and severity of coastal flood events is projected to increase over the next decades, for example due to sea level rise (Taherkhani et al., 2020). Concurrently, the degradation of foreshore vegetation and human-induced subsidence, due to land use and sediment retention by dams, contribute to heightened coastal flood hazards. This presents significant challenges for low-lying coastal communities and ecosystems, which are home to a large portion of the world's population, land area and assets (Bevacqua et al., 2020; Reguero et al., 2015).

The latest IPCC Synthesis Report warns of significant, irreversible damage to coastal areas from climate-induced flooding, with coastal flood hazard continuing to increase well beyond 2100 due to sea level rise (IPCC, 2023). Additionally, exposure to coastal flood events is expected to increase in the future due to factors such as increasing urbanisation in coastal areas (Darlington et al., 2023; Reimann et al. 2023; Neumann et al., 2015).

Addressing coastal flood risk and understanding the potential future impacts requires a comprehensive
understanding of current coastal flood protection measures and standards, both in terms of infrastructure (e.g.,
levees) and nature-based solutions (e.g., mangroves) (Caretta et al. 2022; van Zelst et al., 2021; Toimil et al.,
2020). However, the complexity and challenges involved in quantitatively assessing the level of protection that
existing flood defences provide hinder our understanding of flood protection on a global scale. For example,
challenges arise from the complex interactions between natural (e.g., dunes) and artificial (e.g., dikes) barriers
(Hinkel et al., 2021). Enhanced and detailed data on coastal flood protection is necessary to better prepare for and
mitigate the risks associated with climate change and coastal flooding.

## 2. FLOPROS

In 2016, Scussolini et al. introduced the FLOPROS database, providing the first global collection of information
on flood protection standards across different spatial scales. It consolidates information on protection standards
(expressed as flood return periods) associated with protection measures and regulations. FLOPROS is structured
into three layers: the design layer, which details engineered protection levels of existing river and coastal flood
infrastructure derived from literature; the policy layer, which specifies legislative and normative standards for
protection from river and coastal floods, also derived from literature; and the model layer, which infers river flood
protection standards based on observed relationships with per capita wealth and flood risk.
The FLOPROS model layer assumes a maximum flood protection of a 1000-year return period and a minimum
of a 2-year return period (no protection). An algorithm interpolates these values based on GDP per capita for
different income regions. The model layer determines protection standards for sub-country administrative units
(second level of Nomenclature of territorial units for statistics, or NUTS2 according to Eurostat, 2018) by
calculating expected annual damage and interpolating additional units linearly. This approach overlooks the
complexities of the various physical, socio-economic, and governance factors that influence flood protection
standards, both between and within regions (Klijn et al., 2021).
To our knowledge, FLOPROS, with its coastal update by Tiggeloven et al. (2020), is the only global dataset
documenting existing structural flood protection measures at the sub-national level, making it a cornerstone in
contemporary research endeavours assessing flood risk. Consequently, the database is frequently used in coastal
flood assessments (e.g. Almar et al., 2021; Hermans et al., 2023; Vousdoukas et al., 2018; Yesudian & Dawson,
2021; Ward et al., 2017). FLOPROS was created to support research related to large-scale flood risk management
and has been utilised in several high-level policy documents, including PESETA IV (Feyen et al., 2020), PBL
(2023), and UNEP (2023a, b). Initiatives such as the Intersectoral Model Intercomparison Project (ISIMIP), which
integrates these findings into Integrated Assessment Models like REMIND (Sauer et al., 2021), and webtools such
as Aqueduct Floods also rely on FLOPROS. Additionally, many academic studies assessing current and future
flood risk in coastal areas depend on the FLOPROS database (e.g. Chen et al., 2023; Tiggeloven et al., 2020;
Mortensen et al., 2024; Vousdoukas et al., 2020; Hermans et al., 2023; Devitt et al., 2023; Haasnoot et al., 2021).
As a result, FLOPROS is fundamental to current flood protection assumptions for coastal flood risk and impact
assessments.
However, due to the limitations of FLOPROS, especially the limited number of observations for coastal flood
protection (54 data points from 14 countries), we argue that caution should be exercised in utilising it in coastal
contexts. Overreliance on the dataset may lead to an underestimation of future climate risks, implying protection
where it does not exist, or overestimating adaptation efforts, thus undermining the urgency of climate mitigation.
**3. Current and Future Efforts**
Since the publication of FLOPROS, several initiatives have aimed to improve the representation of flood
protection for coastal regions. FLOPROS mostly contains information on river flood protection in its "design"
and "policy" layers, and exclusively on river flood protection in its "model" layer. Tiggeloven et al. (2020)
extended this by calculating flood protection for coastal regions globally using a comparable model-based
approach but did not include policy or design layers, which may lead to uncertainty. Despite these advancements,
there remains a lack of clear distinction between the use of the original FLOPROS by Scussolini et al. (2016) and
its updated version by Tiggeloven et al. (2020). Frequently, both versions are cited without specifying whether
coastal protection levels are used from the design, policy or model layer (Yesudian & Dawson, 2021). A notable
relevant advancement is openDELve, which compiles an open database referencing the extent and design
specifications of levees for 152 deltas, including levee height, crest width, and construction material, in a
harmonized format (Nienhuis et al., 2022). However, there is a clear contrast in data availability between regions
such as Africa, South-East Asia, and Southern and Central America in comparison to Australia, Europe, UK, and
USA (Nienhuis et al., 2022). Another significant research direction is the detection of flood defence infrastructure
from high-resolution elevation data. Wing et al. (2019) applied a detection algorithm to map levees in the
contiguous U.S., questioning the validity of the wealth-to-protection relationships used in FLOPROS. A similar
method was subsequently applied by Sasaki et al. (2023).
Knowledge of river flood protection standards has been enhanced by studies such as Boulange et al. (2021), which
reflect the protection provided downstream of global hydro dams. In China, river flood protection standards at
higher resolution and confidence levels are available thanks to Wang et al. (2021). Advanced statistical approaches
trained to infer flood protection standards from physical and socio-economic variables have been developed by
Zhao et al. (2023). An indirect approach to infer flood protection standards for Europe, using new data on impacts
and potential flood occurrences, was recently implemented by Paprotny et al. (2024).
**4. COASTPROS-EU: a Coastal Flood Protection Standards Database for Europe**
Despite various advancements in recent years, a dataset with comprehensive global representation of coastal flood
protection measures and their standards is still lacking. We present here, COASTPROS-EU, a new database on
policy standards and defence structures along the European coast (Table S1). The database builds upon the efforts
of FLOPROS and its subsequent improvement by Tiggeloven et al., (2020). However, it differs from FLOPROS
by compiling specifically information on European coastal defences for each NUTS2 region. Furthermore, it
references three typologies of layers, namely geolocated coastal defences, regional coastal defence policies, and
modelled defences based on Tiggeloven et al. (2020). Where applicable, flood protection standards are expressed
in return periods. The "Summary Return Period" summarise the most accurate information layer type regarding
flood protection collected. This column prioritizes the layers type in the following order: (a) geolocated coastal
defences, (b) policy standards, and lastly, (c) modelled defence if no other information is applicable. The overview
of the data availability summary is mapped in Figure 1. The database was produced through two key initiatives.
First, an online survey was distributed within the network of the CoCliCo project (European Union's Horizon
2020 research and innovation programme Coastal Climate Core Services under grant agreement No 101003598)
and the Institute for Environmental Studies (IVM) of Vrije Universiteit Amsterdam. Second, a data workshop was
held at Vrije Universiteit Amsterdam in November 2023, where flood experts collected information on flood
defence and protection standards in their respective language using academic and grey literature (policy reports
and governmental data portals) (Koks and De Plaen, 2023).
The survey consists of an online form targeting flood experts. It collects information related to the scale of the
protection measure, the area protected, the flood protection level expressed in return period and the year of
implementation. In case of a lack of information on physical defences, indication of policy standards applicable
to the area could be filled in with the associated policy measure. Finally, additional data such as geospatial layer
or other relevant information could be uploaded. The information collected were then manually summarised into
the geospatial and policy layers of the database. The survey answers were then archived in the excel file referenced
in the Zenodo repository (De Plaen et al., 2024).

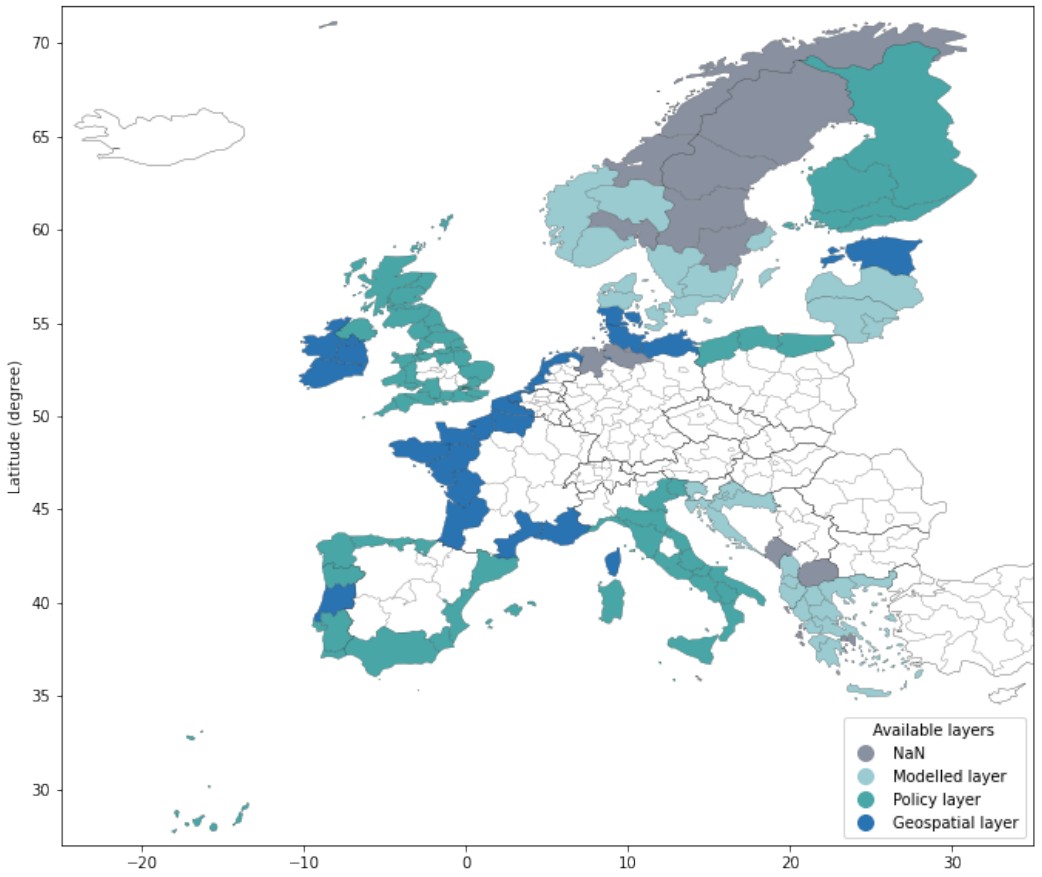

**Figure 1: Data availability overview of COASTPROS-EU, representing the best available coastal protection**
**standards in Europe per NUTS2 region for three typologies of layers: geolocated coastal defences, policy standards,**
**and modeled defence standards.**

Through these combined efforts, we aim to provide a more accurate and comprehensive understanding of coastal
flood protection measures. By incorporating diverse data sources and methodologies, this new database addresses
the critical need for detailed, reliable information to better prepare for and mitigate the risks associated with
climate change and coastal flooding.
While this dataset marks an initial first step, certain limitations must be acknowledged. Firstly, our new dataset is
restricted to Europe and therefore does not meet the need for a global assessment. Moreover, the way protection
levels are quantified and evaluated is critical. Current approaches mainly rely on return periods, providing a
standardized framework suitable for large-scale or regional analyses, with the flexibility to convert between return
periods and defense heights. Yet, incorporating defense heights remains essential, as their significance varies
depending on the specific context and research questions.

### 5. Way Forward: Embracing a Transdisciplinary Community-Based Approach

A global effort is needed to improve data on coastal flood protection, ensuring representation across diverse social
groups, languages, and data-scarce regions. This requires a structured, transdisciplinary approach that integrates
institutional support, community-driven data collection, and multiple sources of information to enhance flood risk
assessments and policy alignment.
Our dataset provides a starting point, but a broader, more structured effort beyond academia is necessary.
Institutional support is key to ensuring sustained data collection and standardized national-level reporting. Cultural
differences in flood protection - such as variations in design standards and their local implementation - must be
captured. While FLOPROS and COASTPROS remain relevant, future efforts must expand coverage, improve
accessibility, and establish globally consistent methodologies.
Earth Observation data remains essential for large-scale assessments of coastal flood protection, offering
standardized insights into infrastructure and nature-based solutions. However, satellite data alone is insufficient.
A multi-method approach integrating satellite-derived information with local expertise, participatory mapping,
and national policy assessments is critical. OpenStreetMap (OSM) provides an opportunity for community-driven
mapping, particularly in regions lacking official records. Aligning these efforts with structured policy reviews and
national reporting frameworks will bridge the gap between local knowledge and institutional decision-making.
Standardized national-level reporting is crucial for improving flood risk assessments and ensuring cross-country
comparability. Systematic reviews of national policies, including flood design standards and their local
application, will enhance data consistency and policy impact. Rather than relying solely on surveys, concrete
recommendations should guide reporting frameworks and promote best practices. By integrating institutional
expertise, satellite observations, and community-driven contributions, we can build a more comprehensive and
equitable approach to flood risk assessment - one that strengthens resilience worldwide.

### Data availability

The excel file and GIS shapefile of COASTPROS-EU are available on the following repository: De Plaen, J. J.-
F. G., Colmenares, M., Koks, E., Scussolini, P., Lena, R., Lincke, D., Kiesel, J., Wolff, C., Tiggeloven, T.,
Peregrina Gonzalez, E. D., Le Cozannet, G., & Sayers, P. (2025). COASTPROS-EU [Data set]. Zenodo.
https://doi.org/10.5281/zenodo.15024139
**Competing interests**
At least one of the (co-)authors is a member of the editorial board of Natural Hazards and Earth System Sciences.

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
