# Peer review of "Brief Communication: Bridging the Data Gap – A Call to"

_Natural Hazards and Earth System Sciences, 2024_

## Author Response (AR1)

Responses RC1

The authors have presented a new database of coastal flood protection measures in Europe, in alignment with FLOPROS. Given the focus on sharing this database through this Brief Communication, I would encourage the authors to include the following, along with some minor comments thereafter:

1. Please further describe the methodology for constructing this database. The only description in this manuscript is the one sentence at Line 104, which leaves a lot of questions about the methodology and its reliability, and the quality of this data. Please comment on the quality of this dataset, and how it compares with that of FLOPROS and its coastal update by Tiggeloven et al. (2020). What further advantages does this dataset provide over the existing dataset?

   Thank you for pointing out this shortcoming. The following was added to clarify the methodology of the survey inquiry: "The survey consisted in online form that was filled in by the flood experts. The survey intended to collect information related to the scale of the protection measure, the area protected, the flood protection level expressed in return period and the year of implementation. In case of a lack of information on physical defenses, indication of policy standards applicable to the area could be filled in with the associated policy measure. Finally, additional data such as geospatial layer or other relevant information could be uploaded. The information collected were then manually summarised into the geospatial and policy layers of the database. The survey answers were then archived in the excel file referenced in the Zenodo repository (De Plaen et al., 2024)."

2. Line 17 - It is prominently mentioned in the abstract that "a new community-driven effort is needed to develop a more complete [coastal flood protection] dataset". However in the remainder of the manuscript, it did not appear that such a community driven effort was employed for developing COASTPROS-EU. Additionally, no information was provided about how such a community driven effort may be implemented.

   We added the following to highlight the collabative effort undertaken to develop the early-stage dataset: "As a first step, we present a early stage dataset compiling COASTtal flood PROtection Standards within EUrope (COASTPROS-EU) elaborated from a survey distributed to flood practictitioners from several european countries."

3. Is COASTPROS an acronym?

   COASTPROS-EU is indeed a acronym. We added: "We propose that a new community-driven effort is needed to develop a more complete dataset, and we present a dataset compiling COASTtal flood PROtection Standards within EUrope (COASTPROS-EU)"

4. Line 37 - The phrase "assessing current flood protection levels" is unclear. Are the authors referring to flood protection measures?

   Thank you for pointing out that the phrasing was unclear. We have changed the

sentence to: "However, the complexity and challenges involved in quantitatively assessing the level of protection that existing flood defenses provide hinder our understanding of flood protection on a global scale". We hope it is more clear now.

5. Line 37 - It would be helpful if the authors included some examples of the complexity and challenges, along with their respective references.

Thank you for the comment, we agree that it would be useful to add an example and have therefore included the following sentence: "For example, challenges arise from the complex interactions between natural (e.g., dunes) and artificial (e.g., dikes) barriers (Hinkel et al., 2021)."

6. Line 51 - What is the difference between sub-country units and "regions" in Line 53?

Thank you for noticing this lack of clarity. Line 51 was editted address this: "An algorithm interpolates these values based on GDP per capita for different income regions. The model layer determines protection standards for sub-country administrative units (second level of Nomenclature of territorial units for statistics; or NUTS2) by calculating expected annual damage and interpolating additional units linearly."

7. Line 52 - What complexities are the authors referring to?

Thank you for pointing out that the complexities are not directly mentioned. We have changed the sentence as follows: "This approach overlooks the complexities of the various physical, socio-economic, and governance factors that influence flood protection standards, both between and within regions." - we hope it is more clear now.

8. Line 54 - The second comma seems unnecessary.

The comma has been removed, thank you.

9. Line 60 - *and* UNEP.

Addressed, thank you.

10. Line 67 - It would be helpful if the authors provided some supporting evidence regarding the "limitations in FLOPROS, especially the limited number of observations for coastal flood protection".

Thank you for this comment. We have added the number of data points in parentheses, and the sentence now reads: "However, due to the limitations in FLOPROS, especially the limited number of observations for coastal flood protection (54 data points from 14 countries), we argue that caution should be exercised in utilising it in coastal contexts."

11. Line 76 - What "new observations" are the authors referring to?

Thank you for pointing out that the phrasinc was unclear. We have changed the

sentence to the following: "Tiggeloven et al. (2020) extended this by calculating flood protection for coastal regions globally using the same model-based approach but did not include policy or design levels, which may lead to a model bias."

12. Line 83 - *and* USA.

Addressed, thank you.

13. Line 98 - What is a NUTS2 region?

Indeed, thank you! NUTS2 has now been defined: "The model layer determines protection standards for sub-country administrative units (second level of Nomenclature of territorial units for statistics; or NUTS2) by calculating expected annual damage and interpolating additional units linearly."

14. Line 99 - The 3 layers appear quite similar to those of FLOPROS. What are the differences between this methodology and FLOPROS?

We have added the following for clarity: " The database builds upon the efforts of FLOPROS and its subsequent improvement by Tiggeloven et al., (2020). However, it differs from FLOPROS by compiling specifically information on European coastal defences for each NUTS2 region. Furthermore, it references three typologies of layers, namely geolocated coastal defences, regional coastal defence policies, and modelled defences based on Tiggeloven et al. (2020)."

15. Line 105 - What is the CoCliCo network?

Thank you for pointing, we have added the following: "First, an online survey was distributed within the network of the CoCliCo project (European Union's Horizon 2020 research and innovation programme Coastal Climate Core Services Horizon 2020under grant agreement No 101003598) project and the Institute for Environmental Studies (IVM) of Vrije Universiteit Amsterdam."

16. Fig 1 - The clarity of the figure could be improved by using more contrasting colors.

The figure colour scheme was adjusted to more contrasting colours – thank you for pointing this out.

Responses RC2

The authors present a new initiative to fill an existing gap on global coastal protection data, and showcasing a new coastal protection dataset for Europe, while making the call for a global joint effort to fill the coastal protection information gap.

The reviewer asks for a few changes and clarifications:

a) it is not clear to the reviewer whether this Brief Communication is intended as (needs clarifying):

a.1 - a call for a global joint effort in building such database, while showcasing an early stage coastal protection database for Europe.

a.2 - a call for a global joint effort in building such database, while presenting a ready-to-use European coastal protection database.

If this is intended as a presentation of an European coastal protection database (a.2), the reviewer asks for clarification on contributions by country in building this database. Europe is a very data dense region in coastal protection and the reviewer is not impressed with the level of detail presented here. If this paper presents an early stage of an European database, please clarify this point in the text.

- Thank you very much for this comment. We have tried to make the intention of our Brief Communication more clear by adding the following:
  - 1) we have changed the title to: "Bridging the Data Gap - A Call to Enhance the Global Representation of Coastal Flood Protection"

  - 2) we have made adjustments to the text in the abstract, clarifying the call to action and the argument for a broader global effort. The new abstract reads as follows: "Abstract. Understanding coastal flood protection is crucial for assessing risks from natural hazards and climate change. However, there is a significant lack of quantitative data on coastal flood protection and their standards, posing a major barrier to risk assessment. FLOPROS, currently the only global database of flood protection standards, relies on limited coastal observations and simplified assumptions. While widely used, this introduces potential uncertainties in impact estimates. To address this gap, we call for a global, community-driven effort to develop a more comprehensive dataset. As a first step, we present a dataset compiling COASTtal flood PROtection Standards within EUrope (COASTPROS-EU), elaborated from a survey distributed to flood practitioners from several European countries, which highlights the need for more extensive and coordinated data collection efforts. Establishing an accurate dataset requires using both bottom-up and top-down approaches and ensuring diverse societal representation."

b) Please expand on methodology.

- Line 123 was added to clarify the methodology of the survey inquirery: " The survey consists of an online form targeting flood experts. It collects information related to the scale of the protection measure, the area protected, the flood protection level expressed

in return period and the year of implementation. In case of a lack of information on physical defences, indication of policy standards applicable to the area could be filled in with the associated policy measure. Finally, additional data such as geospatial layer or other relevant information could be uploaded. The information collected were then manually summarised into the geospatial and policy layers of the database. The survey answers were then archived in the excel file referenced in the Zenodo repository (De Plaen et al., 2024)."

c) Please provide a reference to the Workshop mentioned in lines 106-107.

- Thank you, the reference has now been added.

d) Please add a discussion on the use of Return Periods vs defense heights

- Thank you for the comment! We added the following discussion on return periods and defense heights at line 139: " However, certain limitations must be acknowledged. Firstly, our new dataset is restricted to Europe and therefore does not meet the need for a global assessment. Moreover, the way protection levels are quantified and evaluated is critical. Current approaches mainly rely on return periods, providing a standardized framework suitable for large-scale or regional analyses, with the flexibility to convert between return periods and defense heights. Yet, incorporating defense heights remains essential, as their significance varies depending on the specific context and research questions."

e) Please provide references to all data points in the zenodo database, "grey literature" as well.

- Thank you for pointing this out. However, The brief communication manustript type limits the number of references (https://www.natural-hazards-and-earth-system-sciences.net/about/manuscript_types.html). The references can be found in the survey answers attached to the database.

f) Please introduce the following terms in the text: NUTS2

- NUTS2 has now been define in line 57: "The model layer determines protection standards for sub-country administrative units (second level of Nomenclature of territorial units for statistics, or NUTS2 according to Eurostat, 2018) by calculating expected annual damage and interpolating additional units linearly."

g) Table S1, cited in line 97, is missing

- The Table Supplementary 1 (S1) has been submitted along with the manuscript in a seperate .pdf file as instructed by the journal's guideline. It can be found at the following link: https://nhess.copernicus.org/preprints/nhess-2024-137/nhess-2024-137-supplement.pdf

h) Line 71 needs formatting or joining the main text

- Thank you for pointing this out, Line 71 was fomatted similarly to the rest of the section headings

i) Line 124 Furthermore instead of Further. Comas after the one following Furthermore are not needed.

- This is now addressed, thank you for pointing this out.